# Detection of Feline Coronavirus in Bronchoalveolar Lavage Fluid from Cats with Atypical Lower Airway and Lung Disease: Suspicion of Virus-Associated Pneumonia or Pneumonitis

**DOI:** 10.3390/ani14081219

**Published:** 2024-04-18

**Authors:** Wei-Tao Chang, Pin-Yen Chen, Pei-Ying Lo, Hui-Wen Chen, Chung-Hui Lin

**Affiliations:** 1National Taiwan University Veterinary Hospital, National Taiwan University, Taipei 10672, Taiwan; 2Lab of Small Animal Respiratory and Cardiovascular Medicine, TACS-Alliance Research Center, Taipei, 10672, Taiwan; 3Department of Veterinary Medicine, National Taiwan University, Taipei 10617, Taiwan; 4Animal Resource Center, National Taiwan University, Taipei 10673, Taiwan; 5Graduate Institute of Veterinary Clinical Sciences, School of Veterinary Medicine, National Taiwan University, Taipei 10617, Taiwan

**Keywords:** feline coronavirus, bronchoalveolar lavage, lower airway, lung, feline infectious peritonitis, pneumonia, pneumonitis

## Abstract

**Simple Summary:**

Diagnosing viral pneumonia in small animals before death is uncommon, partly due to the specialized procedure needed to collect lung samples, called bronchoalveolar lavage (BAL), and the infrequent testing for viruses in these samples. Although feline coronavirus (FeCoV) infections are common in cats, it is unclear how often this virus appears in BAL fluid and its relationship with lung problems. This study reviewed 1162 clinical samples submitted for FeCoV testing, of which 25 were BAL samples from cats with atypical lower airway and lung disease. Of the BAL samples tested for FeCoV, 13% (three out of twenty-four) were positive, with no other pathogens detected, suggesting a clinical suspicion of FeCoV-associated pneumonia or pneumonitis. The cats that tested positive for FeCoV in this study appeared to have more abnormal multinucleated cells and nodular lesions in their lungs, but statistical analysis lacked significance, possibly due to the small sample size. An initial corticosteroid treatment yielded improvement of clinical signs in all the cats with suspected FeCoV-associated lung disease, but the long-term prognosis varied. These findings highlight the need for further research on the interplay between FeCoV exposure and lung responses in cats.

**Abstract:**

The premortem understanding of the role of feline coronavirus (FeCoV) in the lungs of cats is limited as viruses are seldom inspected in the bronchoalveolar lavage (BAL) specimens of small animal patients. This study retrospectively analyzed the prevalence of FeCoV in BAL samples from cats with atypical lower airway and lung disease, as well as the clinical characteristics, diagnostic findings, and follow-up information. Of 1162 clinical samples submitted for FeCoV RT-nPCR, 25 were BAL fluid. After excluding 1 case with chronic aspiration, FeCoV was found in 3/24 (13%) BAL specimens, with 2 having immunofluorescence staining confirming the presence of FeCoV within the cytoplasm of alveolar macrophages. The cats with FeCoV in BAL fluid more often had pulmonary nodular lesions (66% vs. 19%, *p* = 0.14) and multinucleated cells on cytology (100% vs. 48%, *p* = 0.22) compared to the cats without, but these differences did not reach statistical significance due to the small sample size. Three cats showed an initial positive response to the corticosteroid treatment based on the clinical signs and radiological findings, but the long-term prognosis varied. The clinical suspicion of FeCoV-associated pneumonia or pneumonitis was raised since no other pathogens were found after extensive investigations. Further studies are warranted to investigate the interaction between FeCoV and lung responses in cats.

## 1. Introduction

Feline coronavirus (FeCoV) infection is fairly common among cats, and most natural infections of enteric FeCoV cause only very mild gastrointestinal signs or remain asymptomatic [1,2]. A small percentage of cats with FeCoV infection develop an effusive or non-effusive form of feline infectious peritonitis (FIP), which leads to vasculitis and pyogranulomatous lesions in multiple organs and can be fatal, especially in young cats [3,4,5]. However, the development of FIP and the extent of lesions are highly variable, depending on the interactions between viral and complicated host factors [1,2]. Moreover, it has been recognized that non-FIP FeCoV can spread systemically in cats, and many non-gastrointestinal tissues, including the lungs, have been identified as sites of persistent infection in cats [1,6].

It has been reported that a high viral load of various respiratory viruses was detected in bronchoalveolar lavage (BAL) samples from immunocompromised human patients with pneumonia [7]. Additionally, the viral etiology was estimated to result in severe pneumonia in approximately one-third of human patients in the intensive care unit, and the presence of the virus in BAL samples was associated with higher mortality [8,9]. While BAL is a diagnostic procedure for various types of lower airway and lung diseases, viral detection is seldom performed in BAL specimens from small animal patients. Most cats and dogs with viral bronchitis and pneumonia are diagnosed based on necropsies, which represents a group of more severe and fatal cases [10,11,12,13,14]. Currently, premortem data on viral pneumonia or lower respiratory tract infection are limited in small animal medicine, partially because of the difficulty in obtaining BAL samples in many clinical cases. This leads to an unclear indication of when the virus should be investigated in BAL samples, as well as the frequency of the presence of FeCoV in such specimens. The association between lower airway/lung abnormalities and the presence of FeCoV in the lung remains unexplored in feline clinical patients without typical FIP.

Considering the current knowledge gap, the objective of this study was to conduct the retrospective analysis of cases with FeCoV investigation in BAL specimens for various reasons. The clinical rationale for inspecting FeCoV in these cases was reviewed from medical records, and the prevalence of FeCoV presence in BAL fluid was reported. Based on the previous literature associated with FIP virus, we hypothesized that the cats with FeCoV detected in the BAL specimens would exhibit more frequent granulomatous inflammation and corresponding clinical characteristics compared to the cats without FeCoV. This study aimed to provide premortem data on FeCoV detection in BAL specimens from a group of clinical feline patients with atypical lower airway/lung disease and propose the clinical suspicion of virus-associated pneumonia or pneumonitis.

## 2. Materials and Methods 

The database of FeCoV reverse transcription-nested polymerase chain reaction (RT-nPCR) from clinical cases at a university teaching hospital was searched for BAL specimens between January 2016 and November 2023. For cases to be included, the BAL fluid had to have undergone thorough cytological and microbiological evaluations, in addition to FeCoV RT-nPCR. If a cat had undergone the BAL procedure more than once, only the results of BAL analysis with FeCoV RT-nPCR performed at the same time point were included.

The BAL samples were collected and evaluated following a standard protocol. The cats were premedicated and anesthetized based on their individual condition. An aseptic technique was used throughout the procedure to prevent oropharyngeal and other potential contaminations, whether using bronchoscopic BAL (B-BAL) or non-bronchoscopic BAL (NB-BAL). For B-BAL, a flexible bronchoscope was introduced either directly through the larynx with care to avoid upper airway contamination or through a sterile endotracheal tube into the trachea. After systematically examining all the accessible bronchi, BAL fluid was obtained from selected bronchi by wedging the scope adequately, instilling preheated sterile saline through the working channel and retrieving the sample by suction or manual aspiration. For NB-BAL, recumbency was determined by the region of interest for sampling. A sterile 6 or 8 French gauge polyvinylchloride tube was passed through a sterile endotracheal tube until it was adequately lodged in the distal bronchus. Preheated sterile saline was instilled into and immediately aspirated. The volume of aliquot instilled was approximately 2 mL/kg, with adjustments made for each case based on the technique or equipment used for sampling and the individual cat’s condition. BAL was performed in at least two sites if the cat was stable enough. The BAL sample was processed within 1 h of collection. The total nucleated cell count was determined on an unfiltered sample, and differential cell counts were determined on a cytospin slide by counting 400–600 cells on average (at least 200) per sample. Giemsa stain was used for routine cytology assessment, and other types of stains were determined according to the initial evaluation findings. Basic microbiological examination included semiquantitative or quantitative aerobic and anaerobic bacterial culture, *Mycoplasma* culture or PCR, and fungal culture. For cases presenting with atypical lower airway and lung disease, additional investigations for other microorganisms were conducted based on suspicion in each case.

Reverse transcription-nested polymerase chain reaction (RT-nPCR) was used to detect feline coronavirus RNA in BAL fluid as previously described [5]. A 3′ untranslated region of the viral genome was targeted by RT-nPCR. In brief, viral RNA was extracted using the PetNAD Nucleic Acid Co-Prep kit (GeneReach, Taichung, Taiwan), and cDNA was synthesized using M-MLV reverse transcriptase and oligo(dT) primer (Invitrogen, Waltham, MA, USA). Subsequently, RT-nPCR was performed using GoTaq reagents (Promega, Madison, WI, USA) and previously described primers [15]. The first PCR was conducted using the forward primer P205 (5′-GGCAACCCGATGTTTAAAACTGG-3′) and reverse primer P211 (5′-CACTAGATCCAGACGTTAGCTC-3′). Nested PCR was conducted using the forward primer P276 (5′-CCGAGGAATTACTGGTCATCGCG-3′) and reverse primer P204 (5′-GCTCTTCCATTGTTGGCTCGTC-3′). A 177-bp PCR product was obtained and analyzed by agarose gel electrophoresis. On the other hand, immunofluorescence antibody staining (IFA) was used to confirm the presence of FeCoV nucleoprotein within the cytoplasm of alveolar macrophage in BAL fluid on a cytospin slide. The slides were prepared concurrently with a routine BAL cytology assessment, fixed with 80% acetone, and then blocked with 10% goat serum for 1 h before proceeding with IFA staining. Mouse monoclonal anti-FeCoV nucleoprotein (Clone FIPV3-70, MCA2194, Bio-Rad, Hercules, CA, USA) at a 1:400 dilution was applied onto the slide and incubated for 1 hr. After washes, the secondary antibody, goat anti-mouse IgG FITC conjugate (Jackson ImmunoResearch, West Grove, PA, USA), at a 1:400 dilution was added onto the slide for another 1 hr. After further washes, the slide was counterstained with DAPI (Invitrogen) and covered with a mounting solution containing DAPI (Vectashield, Newark, CA, USA). The intracellular fluorescence signal was imaged using an automated microscope (Olympus IX83, Tokyo, Japan).

Cases were excluded from final analysis if they presented with any of the following three criteria suggesting oropharyngeal contamination in BAL samples, the presence of *Simonsiella*, the presence of oropharyngeal squamous epithelial cells, or the growth of multiple oropharyngeal flora in bacterial culture of BALF, without other cytological and clinical evidence supporting bacterial bronchitis/pneumonia.

Clinical information was extracted from the medical records of the enrolled cases, including the cat’s age, body weight (BW), gender, breed, clinical signs, blood examinations, diagnostic imaging findings, BAL fluid cytological and microbiological examinations, other clinical evaluations, and final diagnosis. Follow-up information for the cats with positive FeCoV RT-nPCR was obtained from the medical records or from the primary care veterinarian.

Statistical analyses were conducted using SPSS version 26 (IBM Corp, Armonk, NY, USA). A Shapiro–Wilk test was used to determine the normal distribution of continuous variables. Data with a normal distribution were presented as mean and standard deviation (SD), whereas non-normally distributed data were presented as median and range. Mann–Whitney U test was utilized to compare the age, BW, total nucleated cell count (TNCC), and differential counts of BAL fluid between the cats with and without detection of FeCoV, while Fisher’s exact test was used to compare the clinical imaging and specific cytological findings. A *p* value of less than 0.05 was considered statistically significant.

## 3. Results 

A total of 1162 clinical samples were submitted for FeCoV RT-nPCR from January 2016 to November 2023. Among those, 25 were BAL specimens. One cat was excluded from the study due to the suspicion of oropharyngeal contamination in the BAL sample resulting from chronic aspiration. Of the remaining 24 cats, the mean age was 7.9 ± 3.7 years, and the mean body weight was 4.8 ± 1.2 kg. The study population predominantly consisted of male cats (17/24; 71%), with fourteen neutered males, seven neutered females, and three intact males. The majority of cats were domestic shorthair cats (14/24), followed by four American shorthairs (ASHs), and one British shorthair, Persian, ASH-Scottish fold cross, Bengal, Bengal-cross, and Ragdoll-cross cat each.

All these cats had lower airway or lung parenchymal problems. The chief complaints at the initial visit included a cough (23/24; 96%), respiratory distress (13/24; 54%), tachypnea (6/24; 25%), noisy breathing (6/24; 25%), weight loss (5/24; 21%), open-mouth breathing (4/24; 17%), inappetence (4/24; 17%), and decreased activity (3/24; 13%). Thoracic radiography was performed on all 24 cats; the findings included mixed lung patterns (13/24; 54%), bronchointerstitial patterns (11/24; 46%), lung consolidation (8/24; 33%), the presence of nodular lesions (6/24; 25%), lung lobe atelectasis (4/24; 17%), the suspicion of bronchiectasis (3/24; 13%), and lung hyperinflation (3/24; 13%). At the time of BAL, none of the cats exhibited blood examination abnormalities typically associated with systemic FIP, such as hyperglobulinemia. Additionally, all the cats were absent of ascites and pleural effusion at this time point.

The decision to incorporate FeCoV RT-nPCR as one of the laboratory tests for the BAL specimens in these cases was based on various suspicions, including atypical clinical presentations that did not align with typical feline lower airway disease (FLAD) or aspiration pneumonia (19/24; 79%), the presence of multinucleated cells in BAL cytology (13/24; 54%), nodular lesions or granulomas observed on the thoracic computed tomography (CT) images (6/24; 25%), an extensive pathogen search due to unresolved pneumonia of uncertain cause (3/24; 13%), and the suspicion of FIP by the primary care veterinarian despite unclear reasons (2/24; 8%).

The FeCoV RT-nPCR testing of the BAL samples from these 24 cats revealed a 13% (3/24) positivity rate, with no other bacterial, fungal, or viral pathogens identified in these three cases. Of the three cats with positive RT-nPCR results for FeCoV, two cats also had a positive IFA result, confirming the presence of FeCoV within the cytoplasm of alveolar macrophages in the BAL fluid (Figure 1). These three cats were diagnosed with suspected FeCoV-associated lung disease. All three cats presented with multinucleated cells based on BAL cytology (Figure 1), and two cats presented with nodular lesions on the thoracic CT images (Figure 2).

The microbiological investigations of twenty-one cats with negative FeCoV RT-nPCR results discovered various bacterial, fungal, and viral microorganisms: four with the molecular detection of fungal organisms not known for their role as pathogens; two with clinically relevant bacterial growth in the culture (1.7 × 10^3^ colony-forming units per milliliter of BAL fluid), including *Pasteurella multocida* plus *Pasteurella dagmatis* and *Haemophilus felis*; one with a positive PCR for *Mycoplasma* spp.; one with the growth of *Cryptococcus neoformans* in the culture; and one with a positive PCR for feline herpesvirus.

The clinical characteristics and diagnostic findings among the cats with and without detection of FeCoV were compared. There were no significant differences in age, BW, TNCC, and differential counts of BAL fluid between the cats with and without detection of FeCoV (Table 1). In the cats with FeCoV present in BAL specimens, nodular lesions (66% vs. 19%, *p* = 0.14) and multinucleated cells (100% vs. 48%, *p* = 0.22) were more frequently found on the CT imaging and BAL cytology, respectively, but these differences did not reach statistical significance.

Follow-up information was available for all three cats that tested positive for FeCoV RT-nPCR in the BAL fluid. Since no pathogen other than FeCoV was identified after an extensive survey, all three cats received a steroid treatment, either oral prednisolone or a combination of oral prednisolone and inhaled corticosteroid. All three cats exhibited a positive response to the steroid treatment, showing significant improvement in both the clinical signs and radiological findings. At the time of writing, one cat remained free of respiratory signs 16 months after BAL with an inhaled corticosteroid treatment. Another cat showed a gradual worsening in clinical signs despite the initial improvement. This cat developed a scant amount of pleural fluid and passed away 32 months after BAL due to respiratory distress. The last cat’s clinical signs acutely exacerbated after a stressful event (a visitor staying at home during the holidays). Subsequently, the cat developed a significant amount of pleural effusion consistent with modified transudate, characterized by a high proteinaceous content. The FeCoV PCR of pleural effusion was not performed by the primary care clinician who conducted thoracentesis. This cat passed away 4 months after BAL.

## 4. Discussion

The present study revealed that FeCoV was present in 13% of the BAL specimens from cats showing atypical lower airway or lung abnormalities. All the cats with a positive FeCoV RT-nPCR result in the BAL fluid were found to have multinucleated cells based on cytology, and two out of three cats presented with nodular lesions on the thoracic CT images. Corticosteroid administration demonstrated initial positive treatment responses in all three cats with suspected FeCoV-associated lung disease, but carried a variable long-term prognosis.

Among the clinical feline patients with infectious pneumonia, bacteria are typically the most common cause, yet fungal, parasitic, protozoal, and viral etiologies are also potential pathogens to consider [11,16]. In a postmortem study, FIP was the most common viral etiology of infectious pneumonia in cats, responsible for nine of the eleven cases with viral pneumonia [11]. Conversely, FeCoV was absent in all the formalin-fixed and paraffin-wax-embedded lung samples in another more recent study involving 69 cats with severe and lethal pneumonia [14]. Therefore, FeCoV seems to be variably present among the known lethal cases of feline pneumonia across different populations. Alternatively, at present, FeCoV may cause lethal pneumonia less often compared to its prevalence in earlier years; however, further studies are needed to examine this hypothesis.

The typical histopathological lesions of FIP consist of vasculitis, perivascular necrosis, pyogranulomatous, and fibrinous lesions in the liver, kidneys, spleen, intestines, mesenteric lymph nodes, peritoneal lining, pleura, central nervous system, and any other affected organs [1,2,10]. Although it is less common, the lungs can also be involved and described as having pyogranulomatous pneumonia, fibrinopurulent pneumonia, interstitial pneumonia, or fibrinonecrotic pleuropneumonia [1,10,11,12]. Nevertheless, most of these data were collected from life-threatening and lethal FIP cases, whereas little is known about clinical cases of the non-effusive type with the lungs as the primarily affected organ. In our study, all three cats with FeCoV in the BAL fluid exhibited multinucleated cells based on cytology, and two cats showed nodular lesions based on thoracic CT. Although the presence of multinucleated cells or nodular lesions was not statistically different between the cats with positive and negative FeCoV RT-nPCR results in the BAL fluid, these cytological and imaging features still imply the possibility of pulmonary granulomatous lesions associated with FIP in these cats [10,17], especially when no other etiologies could be identified.

A debate arising from the discovery of three cats with only FeCoV present as the sole microorganism in their lungs is whether this represents an uncommon manifestation of non-effusive FIP. While typical FIP primarily affects young cats, the non-effusive type can exhibit a more chronic and insidious course, posing greater challenges in diagnosis [1,2,4]. Furthermore, although the disease process of FIP is fundamentally characterized by a systemic inflammatory response, there have been occasional reports of cases with “focal FIP,” where the lesions are restricted to specific areas such as the intestines or mesenteric lymph nodes [1,4,18,19]. Without histopathological evidence, it is challenging to definitively conclude that the three cases in our study represent a pulmonary form of focal FIP. This is a commonly unavoidable challenge in non-fatal cases, particularly when the lung lesions are multifocally distributed in multiple lobes, and thus not likely to benefit from surgical lobectomy.

Another plausible hypothesis for the lung lesions in these cats is an immune-mediated pneumonitis triggered by non-FIP FeCoV. Various viruses, including coronavirus, have been reported to induce an excessive host immune response, cytokine production, and secondary organizing pneumonia, which can respond rapidly to steroid treatment [20,21,22,23]. It is now well known that cats can carry non-FIP FeCoV systemically [2,6,24], but there is still limited understanding of all the potential consequences in the lungs following exposure to FeCoV other than FIP.

Two out of the three cats with FeCoV detected by RT-nPCR in our study also yielded positive results for IFA. The remaining cat represents the first case in which we obtained a positive FeCoV RT-nPCR result from BALF. Consequently, IFA was not promptly conducted on a fresh sample consisting of live cells, resulting in an invalid result. The detection of FeCoV by RT-PCR has been utilized in various clinical samples with the suspicion of FIP, although it is not specific for FIP, as positive FeCoV RNA could be found in the tissues from cats without FIP [25]. Conversely, while the immunostaining of FeCoV antigens in the macrophages of affected tissues is generally considered specific and reliable, the variable sensitivity of various technical methodologies across different sample types has been reported [2,25]. All these methods should be viewed as supportive evidence for the presence of FeCoV rather than providing a definitive diagnosis of FIP [2].

Survival in cats with FIP is relatively short in acute cases or the effusive type, ranging from several days to weeks [1,26]. While the disease course of the non-effusive type could be protracted and likely more variable [4,26], previous cases with focal FIP were mostly euthanized or succumbed within one year of histopathological diagnosis in an earlier study [18]. In comparison to our study, two out of three cats with suspected FeCoV-associated pneumonia or pneumonitis remained alive one year after the detection of FeCoV in the BAL fluid. The long-term outcomes of cats with non-FIP FeCoV infections are uncertain, as it is unclear whether these cats may later develop FIP or other diseases [24]. Interestingly, two cats in our study eventually developed pleural effusion, with one presenting a sufficient amount for thoracentesis. Unfortunately, the effusion obtained by the primary care clinician at the time was not examined for FeCoV; thus, a definitive diagnosis of effusive type FIP cannot be concluded in this case. However, the pleural effusion of this case was characterized as a modified transudate with a high proteinaceous content, aligning with the typical effusions observed in cats diagnosed with FIP [2,27]. These scenarios resemble those cases initially presenting with non-effusive FIP and subsequently developing effusions at a later stage [2].

Corticosteroids may offer palliative relief by suppressing or modulating the inappropriate immune response in FIP cases without providing a cure, and it is potentially useful for cats with lesions confined to a single tissue [4,26]. In cases of organizing pneumonia secondary to viral exposure or infection, corticosteroid therapy is the first-line choice and is associated with a rapid response in human patients [20,21,23]. Consequently, prednisolone was prescribed in the three cases with the suspicion of FeCoV-associated pneumonia or pneumonitis in our study. Although an initial therapeutic response was observed in all the cats, the long-term prognosis varied significantly. These follow-up conditions suggest that the corticosteroid treatment positively impacted the improvement of clinical signs, but did not effectively alter the underlying disease process. Currently, the emerging evidence supports the use of nucleoside analogs as effective antiviral agents for treating FIP [25,28,29,30,31,32], but these drugs were not available in the authors’ country at the time of diagnosis of the two cats developing pleural effusion. One cat is still alive at the time of writing, but free of clinical signs after corticosteroid therapy. Further research is required to assess whether nucleoside analogs would benefit cases of FeCoV-associated lung disease. While this treatment option might be considered in the future, the prerequisite is to consider the suspicion of FeCoV-associated pneumonia or pneumonitis as a potential differential diagnosis.

This study has several limitations. Firstly, being retrospective in nature, the diagnostic workup lacked standardization. While all the cats underwent basic diagnostic procedures such as thoracic radiography, not all underwent computed tomography, potentially resulting in overlooked small nodular lesions. Secondly, the decision to perform FeCoV RT-nPCR on the BAL specimen was made by the attending clinicians, potentially introducing a selection bias. Thirdly, despite the collection of cases spanning over a 7-year period, the small sample size, particularly the limited number of cats with positive FeCoV RT-nPCR results, may have compromised the statistical power, affecting the ability to detect significant differences between the groups. However, the small sample size reflects the under-investigated nature of this situation. The results of this study may raise awareness among clinicians and researchers about the possibility of a viral etiology, with the hope that more data will be gathered in the future. Lastly, the absence of histopathology from the two FeCoV-positive cats that died precludes the establishment of a definitive diagnosis of FIP.

## 5. Conclusions

Viral pathogens have rarely been investigated in the BAL specimens of small animal patients, and the present study revealed that FeCoV was detected in 13% of cats exhibiting atypical lower airway or lung abnormalities. In cases with unexplained clinical findings, the presence of multinucleated cells on BAL cytology, or nodular lesions on thoracic images, the suspicion of FeCoV-associated pneumonia or pneumonitis should be taken into consideration. The premortem identification of FeCoV within the alveolar macrophages in clinical cases can be achieved through BAL with RT-nPCR and IFA, particularly when the typical pathogens are not found after thorough investigation. The cats with FeCoV-associated lung disease had variable prognoses. Corticosteroid treatments contributed to the improvement of clinical signs in all three cats with suspected FeCoV-associated pneumonia or pneumonitis in our study, but two cats eventually developed pleural effusion prior to death. When FeCoV is detected in the BAL fluid of cats with granulomatous lung lesions, either a pulmonary form of focal FIP or an immune-mediated pneumonitis triggered by FeCoV should be considered. Further studies are warranted to investigate the interaction between virus exposure and the host response in the lungs of cats.

## Figures and Tables

**Figure 1 animals-14-01219-f001:**
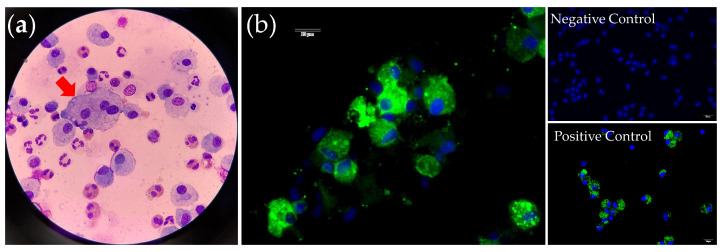
(**a**) BAL cytology from a cat with positive RT-nPCR result for FeCoV in the BAL fluid. Giant trinucleated alveolar macrophages (arrow) were noted, with mixed neutrophilic, eosinophilic, and histiocytic inflammation. Giemsa stain. (**b**) IFA demonstrated the presence of FeCoV nucleoprotein within the cytoplasm of alveolar macrophages in a cat with positive RT-nPCR for FeCoV in the BAL fluid. An FIP pleural effusion sample slide was used as a positive control, and a non-FIP BAL sample slide was used as a negative control. Scale bar: 20 μm.

**Figure 2 animals-14-01219-f002:**
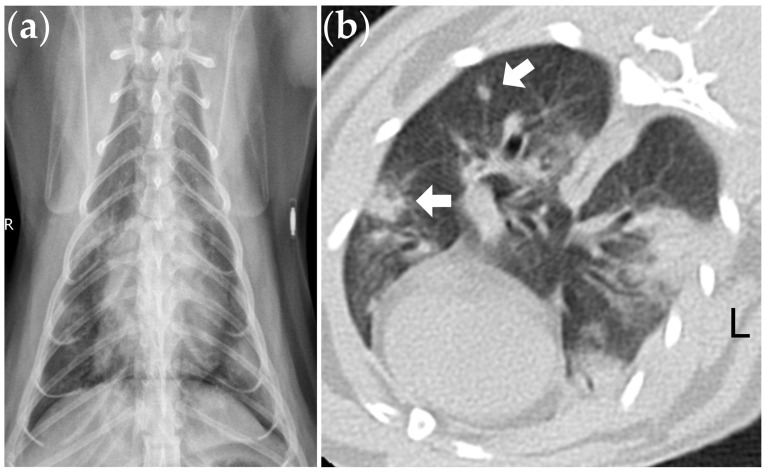
(**a**) Thoracic radiography from a cat with positive RT-nPCR result for FeCoV in the BAL fluid, showing multifocal distribution of abnormal opacities involving more than one lung lobe. (**b**) Lung high-resolution CT of the same cat revealed multifocal consolidation lesions and nodular lesions (arrow). The CT scanning was performed under sedation without general anesthesia (window width: 1500 HU; window level: −400 HU).

**Table 1 animals-14-01219-t001:** Signalment, BAL cytology, and imaging findings in cats with positive and negative FeCoV RT-nPCR results. Data are presented as median with range, or percentage with numbers.

	FeCoV Positive	FeCoV Negative	*p* Value
Age (years)	10.0 (3.0–10.0)	8.0 (1.0–15.0)	0.87
BW (kg)	3.4 (3.2–5.2)	4.7 (2.8–7.0)	0.20
TNCC (cells/μL)	984 (851–1120)	895 (287–5099)	0.87
Neutrophils (%)	27.3 (9.0–31.7)	45.0 (2.6–88.5)	0.40
Eosinophils (%)	29.5 (25.0–49.5)	6.0 (0.1–80.6)	0.20
Lymphocytes (%)	5.9 (2.8–13.8)	4.1 (0.6–28.9)	0.90
Macrophages (%)	36.1 (27.8–41.8)	22.8 (7.4–62.3)	0.27
Multinucleated cells in BAL fluid (%)	100 (3/3)	48 (10/21)	0.22
Nodular lesions on CT image (%)	66 (2/3)	19 (4/21)	0.14

BAL; bronchoalveolar lavage; BW, body weight; CT, computed tomography; TNCC, total nucleated cell counts.

## Data Availability

The data presented in this study are available upon request from the corresponding author.

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
