# Peer review of "Detection of Feline Coronavirus in Bronchoalveolar Lavage Fluid from Cats with Atypical Lower Airway and Lung Disease: Suspicion of Virus-Associated Pneumonia or Pneumonitis"

_animals, 2024, doi:10.3390/ani14081219_

Round 1

Reviewer 1 Report

Comments and Suggestions for Authors

Feline coronavirus (FeCoV) infection is a common and mild cat disease; however, it can develop into a more severe and fatal condition known as feline infectious peritonitis (FIP).

In this manuscript conducted by Chang et al., 25 bronchoalveolar lavage (BAL) samples were examined for the presence of FeCoV and other pathogens utilizing various diagnostic techniques. Using nested PCR specifically targeting FeCoV, three out of the 25 samples tested positive. Cytological analysis revealed the presence of multinucleated cells in these three positive samples, along with positive immunostaining with antibody binding to FeCoV nucleocapsid protein indicating the presence of the virus in two of these samples. Additionally, nodular lesions were observed in the lungs of two of these cats.  

The study design appears to be logical, and the data provided offer support for the conclusions drawn. Nevertheless, there are major concerns regarding the clarity of the study’s objectives and the presentation of the data. It is essential to provide more comprehensive explanations regarding the purpose of the study and to present the data in a more logical and statistical manner. These concerns have been outlined for further revision.

1)      The authors did not adequately explain the importance or significance of this study. While they briefly mentioned that “This (limited data on viral pneumonia) leads to an unclear when the virus should be investigated in BAL samples, as well as the frequency of the presence of FeCoV in such specimens.” However, this rationale lacks sufficient persuasiveness for readers.

A.         The main concern in this study is clinical relevance in Detecting FeCoV in BAL. First, BAL is a very limited technique in small animals with pneumonia or respiratory problems. Second, once FeCoV develops into FIP, it becomes a systemic infection. Other sample types, such as blood and ascites, may be more suitable for FeCoV diagnosis and monitoring disease progression. Thus, the authors must provide a more convincing explanation to persuade readers of the importance of BAL in studying FeCoV-associated pneumonia.  

2)      In Fig 1, the authors present a single microscopy image and claim that 2 or 3 PCR-positive cats have giant cells (Multinucleated cells) or IFA-positive cells. However, their presentation needs improvements.

A.         Concerning the presence of giant cells (Fig 1a):  

                                         i.               It is recommended that they analyze images of multiple sights of slides from each cat and quantify the number of giant cells relative to total cells. Subsequently, they should compare the percentage of giant cells between FeCoV-negative and positive cats. This approach would render the data statistically robust.  

                                       ii.               it remains unclear whether the formation of giant cells is directly attributed to FeCoV. To substantiate their claim, the authors should present evidence of nucleocapsid protein-positive giant cells rather than relying solely on Giemsa staining.

B.         Concerning the IFA staining results (Fig 1b):

                                         i.               It is crucial for the authors to include appropriate controls, both negative and positive, to ensure the reliability and interpretation of the results.  Without the controls, the significance of the image is compromised.

                                       ii.               Similar to their approach for giant cells, the author should conduct a quantitative analysis by counting and calculating the percent of nucleocapsid protein (N protein) positive cells, followed by a comparison between FeCoV-negative and positive cases.

                                     iii.               There is a lack of evidence supporting the assertion that the stained cells are macrophages. To address this uncertainty, the authors should consider labeling cells with surface markers and analyzing them using flow cytometry or confocal microscopy to accurately identify and characterize the cell types involved. This approach would enhance the credibility and comprehensiveness of their findings.  

3)      Concerning the radiology results (Fig 2):

A.         A Similar issue to Fig 1 arises. While the image effectively depicts the multifocal distribution of abnormal opacities in multiple lung lobes in positive cats, it remains unclear whether nodular lesions presented in positive cats were associated with FeCoV infection. To address this ambiguity, I recommend that the authors quantify the number or size of the lesion in each individual cat and compare these findings with those of normal or healthy cats (negative control). Subsequently, they should establish statistical relevance between radiological findings and FeCoV-positive results.

4)      In Table 1, the authors compared the percent of multinucleated cell-positive cats between FeCoV positive and negative populations (100% vs 48%). Similarly, they compared nodular lesions-positive cats between FeCoV-positive and negative populations. However, this comparison may not be appropriate.

A.         As previously mentioned, the authors should calculate the percentage of multinucleated cells in each individual cat and calculate the average percentage of multinucleated cells in FeCoV positive and negative populations separately. Subsequently, they should compare these two values. This approach will enable the identification of potential statistical differences between the two groups.

5)      Did the authors test for FeCoV in the sera or other samples collected from the cats undergoing BAL? If so, they should present the result. For example, they should present the positive rate in BAL fluid vs. serum samples. If the number of double-positive cases is higher, BAL could serve as an alternative method for testing FeCoV or FIP.

6)      The author employed RT-nested PCR for the detection of FeCoV RNA, a technique developed three decades ago. Was there a specific rationale for opting for this older method? It is noteworthy that while nested PCR is highly sensitive, it is also prone to generating false-positive results. Were DNA sequencing analyses conducted on the amplified DNA to validate the finding? Currently, Reverse transcription real-time PCR(RT-qPCR) is favored for its enhanced specificity and the capability to quantitatively assess the target gene. Therefore, I would suggest considering the adoption of this to diagnose FeCoV.  

Author Response

Please see the attachment: Cover letter and Author response (Manuscript ID animals-2922523).pdf

Reviewer 2 Report

Comments and Suggestions for Authors

This is a novel and interesting study of the presence of FeCoV in tissue.  Very little is known about FeCoV related pathology in cats outside of the gastrointestinal cat.  The sample size is small and the results fail to reach statistical significance, but this is a common problem in veterinary clinical research. The novelty of the information about the presence of FeCoV in BAL samples in the study population makes at an interesting and valuable contribution to the field of feline coronavirus research.  

My main suggestion to the authors is to include information on what gene is targeted by the nested PCR sequences.  It may also be interesting to include a citation that compares the sensitivity of RT-PCR to ICH in detecting FCoV, if available.  Lastly, to enhance the utility of the data collected, it would be interesting to know what other pathogens were identified in the BAL samples and whether any co-infections were detected. The authors highlight that FeCoV was the only pathogen detected in the positive BAL samples this study, but it may be of interest to the readers to have a sense of the prevalence of other viral and bacterial infections.  

Author Response

(The authors gave the same response as above.)

Round 2

Reviewer 1 Report

Comments and Suggestions for Authors

The authors' responses adequately address most of the questions and suggestions raised.